# Effects of SARS-CoV-2 on Pulmonary Function and Muscle Strength Testing in Military Subjects According to the Period of Infection: Cross-Sectional Study

**DOI:** 10.3390/diagnostics13101679

**Published:** 2023-05-09

**Authors:** Josuel Ora, Paola Rogliani, Federica Ferron, Marilisa Vignuoli, Letizia Valentino, Giancarlo Pontoni, Francesca Di Ciuccio, Roberto Ferrara, Tommaso Sciarra

**Affiliations:** 1Respiratory Medicine, Policlinico Tor Vergata Foundation, 00133 Rome, Italy; 2Respiratory Medicine, Department of Experimental Medicine, Tor Vergata University, 00133 Rome, Italy; 3Physical Medicine and Rehabilitation Unit, Italian Army Medical Hospital, 00184 Rome, Italy; 4Physchiatry Section, Phychophysiological Selection Office, Italian Army National Recruitment and Selection Center, 06034 Foligno, Italy

**Keywords:** COVID-19, pulmonary function test, 6MWT, hand grip, exercise

## Abstract

Background: Pulmonary function can be impaired as a long-term consequence of SARS-CoV-2 infection. The aim of this study was to evaluate the effect of SARS-CoV-2 infection on pulmonary function, exercise tolerance, and muscle strength in healthy middle-aged military outpatients according during the period of infection. Methods: A cross-sectional study was carried out from March 2020 to November 2022 at the Military Hospital “Celio” (Rome, Italy). If someone had a diagnosis of SARS-CoV-2 infection certified by molecular nasal swab and if they performed pulmonary function tests, diffusion of carbon monoxide (DL’co), a six Minute Walk Test (6MWT), a Handgrip (HG) Test, and a One Minute Sit to Stand Test (1′STST). The included subjects were divided into two groups, A and B, according to the period of infection: A) from March 2020 to August 2021 and B) from September 2021 to October 2022. Results: One hundred fifty-three subjects were included in the study: 79 in Group A and 74 in Group B. Although the values were within the normal range, Group A had smaller FVC, FEV_1,_ and DL’co compared to Group B. Group A also walked a shorter distance at the 6MWT and performed fewer repetitions in the 1′STS test compared to Group B. In both groups, the DL’co (%predicted) correlated with the 6MWT distance (R^2^ = 0.107, *p* < 0.001), the number of repetitions of the 1′STST (R^2^ = 0.086, *p* = 0.001), and the strength at the HG test (R^2^ = 0.08, *p* < 0.001). Conclusions: This study shows that the SARS-CoV-2 infection in healthy middle-aged military outpatients was more severe in the first waves than in the later ones and that, in healthy and physically fit individuals, even a marginal reduction in resting respiratory test values can have a major impact on exercise tolerance and muscles strength. Moreover, it shows that those infected more recently had symptoms related to the upper respiratory tract infection compared to those of the first waves.

## 1. Introduction

Coronavirus disease 2019 (COVID-19) is a disease caused by severe acute respiratory syndrome coronavirus 2 (SARS-CoV-2) that can clinically present itself as a broad variety of symptoms from mild illness to a serious disease leading to hospitalization [1].

So far, many countries have experienced several waves of the disease related to different circulating variants named using the Greek alphabet proposed by the World Health Organization (WHO) [2]. The main variants were: the Alpha variant (alias of B.1.1.7), first detected in the United Kingdom by the end of September 2020; the Beta variant (alias of B.1.351), which emerged independently in South Africa in May 2020; the Gamma variant (alias of P.1), first identified in Brazilian travelers in Japan; the Delta variant (alias of B.1.617.2) first detected in India in May 2021; the Omicron (alias of B.1.1.529) variant in late November 2021 in Botswana and South Africa.

In Italy, three different variants have been predominant: the Alpha (December 2020–July 2021), the Delta (June 2021–September 2021), and the Omicron variant (October 2021–January 2023) [3], and although five different waves have been identified, the first three occurred between Dec 2020 and the summer of 2021 corresponding to the variant Alpha, Beta, Gamma and Delta and the latter two between Fall 2021 and the present [3].

Another turning point in the evolution of the epidemic was vaccination. In Italy, three different vaccines (Pfizer BNT162b2, Moderna mRNA-1273, Janssen Ad26.COV2.S) have been approved, and based on data from the Italian Ministry of Health covering more than 90.24% of the over 12 population. Interestingly, neutralizing antibodies induced by all primary vaccine regimens showed little cross-reactivity with Omicron [4], although T-cell responses induced by vaccines have very good (>80%) cross-reactivity to Omicron and prior variants [5].

Nowadays, although the epidemic seems to be in its descendent phase, the consequences of the infection are becoming more dominant, and this syndrome is called “long-COVID” [6]. Symptoms of COVID-19 can appear 2–14 days after exposure to the virus and may range from mild to severe, although some subjects may be asymptomatic. The most common symptoms are fever, cough, and tiredness, but many other symptoms may also be present, such as shortness of breath or difficulty breathing, muscle pain, headache, loss of taste or smell, sore throat, runny nose, nausea or vomiting, diarrhea [6]. Nevertheless, symptoms can differ according to the viral variant, immune state, and vaccination [7]. As COVID-19 is a multi-systemic disease, complications of the infection may involve any system, although the involvement of the respiratory tract is predominant. The acute infection can last 2–3 weeks depending on clinical severity presentation; however, many patients may present symptoms lasting more than 12 weeks [8,9,10,11]. This syndrome is defined differently as Long-COVID Syndrome (‘Long-COVID’ in the United Kingdom or ‘long-haulers’ in the United States), post-acute COVID-19 syndrome (PACS), or post-COVID-19 syndrome (PCS) [6]. In a pooled analysis of the prevalence of post-COVID symptoms, including 27 papers, Aiyegbushi et al. [8] found that the most prevalent reported symptoms were fatigue 47% and dyspnea (shortness of breath) 32%.

Some studies have already shown some pathological abnormal lung function in COVID-19 survivors, characterized by altered lung diffusion capacity of carbon monoxide (DL’co), but not the DL’co/Alveolar Ventilation (VA) and a restrictive ventilatory defect [12,13,14,15,16]. We have already demonstrated that in 75 post-COVID survivors, PFTs were within the normal range, although after six months, a mild restrictive spirometric pattern could be detected just in very severe subjects [16]. Moreover, the only persistent pathological sequelae of SARS-CoV-2 infection were a mild desaturation at 6MWT, unrelated to the severity of the infection [16,17].

Those data are mostly referred to the first waves of infection, and maybe they could not be representative of the evolution of the virus. So far, it is not clear if the new variants (mostly Omicron) have a different effect on pulmonary function tests [18].

The aim of this study was to compare the different effects of SARS-CoV-2 infection on pulmonary function tests and exercise tolerance in healthy subjects according to the period of infection.

## 2. Materials and Methods

This was a retrospective cross-sectional study performed at the Military Hospital “Celio” (Rome, Italy) and was conducted according to the STROBE guidelines [19,20] (checklist in the Appendix A). Subjects were recruited if they had a diagnosis of SARS-CoV-2 infection certified by molecular nasal swab and if they performed pulmonary function tests (PFTs), 6MWT, Handgrip test (HGT), and 1 min Sit to Stand test (1′STST). All subjects belonged to the four-Armed Forces (Italian Army, Carabinieri, Navy, Air Force) and were eligible for Unconditional Military Service. Subjects were excluded if they were unable to perform functional motor and/or respiratory tests with adequate cooperation, if they were not eligible for Unconditional Military Service, or if there were missing data.

All subjects were evaluated after the SARS-CoV-2 infection by a special team composed of a pulmonologist, a sports doctor, a physiatrist, and a physiotherapist.

Subjects were divided into two groups according to the time of infection: Group A if subjects were infected before September 2021 and Group B if they were infected after September 2021. Although the decision was arbitrary, based on epidemiological data of the occurrence of the various waves, it reflects the probability of Omicron infection of Group B [3] and the vaccination trend of the study population. However, molecular analysis of variants has not been recorded as it is not routinely performed. The investigations were carried out following the rules of the Declaration of Helsinki of 1975, revised in 2013. This study was approved by the ethics board of “Celio” Rome (n. CE/2022u/04/e-31/12/2022).

### 2.1. Pulmonary Function Testing

Complete PFTs, including forced vital capacity (FVC) and DL’co, determined with the single breath technique (SB), were carried out according to the America Thoracic Society (ATS) guidelines [21] on a Quark PFT (Cosmed, Roma, Italy) using ERS 93 extended reference values [22].

### 2.2. Six-Minute Walk TEST (6MWT)

The 6MWT was performed in a 25 m, straight indoor hallway according to ATS guidelines [23]. All patients were tested under standardized conditions by trained operators. Heart rate and oxygen saturation were measured continuously at rest (baseline) and during the test until recovery. The Chekme Pro was used to measure oxygen levels during the test (Viatom Technology Co., Shenzhen, Cina). The 6MWT distance was expressed both as an absolute value in meters and as a %predicted value (%pr) using the Enright and Sherill equations [24].

### 2.3. Handgrip Test (HGT)

The handgrip test was performed according to ‘The American Society of Hand Therapists’ protocol [25]. Grip strength was measured in a seated position, with the shoulder abducted, elbow flexed at 90 degrees, and forearm in a neutral position. We selected the mean value of three maximum grip efforts of both dominant and non-dominant hands, and we evaluated the results according to male grip-strength values and female grip-strength values [26].

### 2.4. Minute Sit-to-Stand Test (1′STST)

All 1′STSTs were performed according to a standardized protocol by trained staff [27]. We used a standard chair (height 46–48 cm) with a flat seat and no armrests, stabilized against a wall. Patients were asked to sit with their legs hip-width apart and flexed to 90°, with their hands stationary on the hips without using the hands or arms to assist movement. They were instructed to stand completely straight and touch the chair with their bottom when sitting, but they need not sit fully back on the chair. Patients were monitored during the test; in particular, we recorded: heart rate (HR), SpO_2_, the number of repetitions, and the rating of perceived exertion (RPE) for dyspnea and leg fatigue (using Borg Scale), before and after the test.

### 2.5. Chest CT Scan

Chest CT scans were performed only when clinically necessary and given a qualitative score as follows: Score 0: normal lung; Score 1: ground glass opacity (GG); Score 2: crazy paving pattern or GG opacity with consolidation; Score 3: diffuse consolidation [28].

### 2.6. Statistical Analysis

Statistical analysis was carried out employing Microsoft Excel software. Group comparisons were made using Student’s *t*-test or two-way analysis of variance (ANOVA) as appropriate. For categorical variables, the chi-square test was used. A *p*-value < 0.05 was considered statistically significant. Univariate correlations were examined using Pearson’s product-moment correlation. Data are presented as average ± standard deviation (SD) if not differently specified.

## 3. Result

### 3.1. Subjects

Subjects’ characteristics are shown in Table 1. Of the 170 subjects evaluated, 5 were excluded due to lack of PFR and 12 due to lack of follow-up. One hundred fifty-three subjects were included in the study: 79 were infected before September 2021 (Group A), and 74 were infected after September 2021 (Group B). A total of 16 subjects were not included in the analysis because either the date of infection (12) or the PFT (4) was unavailable.

The mean follow-up was six months (140 ± 88 days), longer in group A (40 days, *p* = 0.006) mainly because these were the first cases to occur. The main difference between the two groups was the number of hospitalized patients (in Group A, 72% of patients were hospitalized compared to 12.2% of patients in Group B, *p* < 0.01), the severity of the infection evaluated by the need for oxygen supplementation or high flow nasal oxygen or non-invasive mechanical ventilation (Table 1). Comorbidities are reported in Table 1. In Group A, systemic arterial hypertension was more prevalent than in Group B. On the other hand, metabolic disorders were more common in Group B compared to Group A.

Vaccination status was different between the two groups. In Group A, only three subjects had been vaccinated, while in Group B, 66 subjects had more than one dose of the vaccine, one subject had a single dose of vaccine, and two had a previous SARS-CoV-2 infection. Four subjects in the group were not vaccinated.

A CT scan was performed only in more severe subjects when clinically necessary.

A total of 59 subjects had chest CT scans, 42 in Group A and 17 in Group B. In Group A, 4 subjects had a normal CT scan, 19 showed GG opacity, 9 GG opacity and consolidation, and 10 diffuse consolidations. In Group B, none had a normal CT scan, 19 presented GG opacity, 9 GG opacity and consolidation, and 10 diffuse consolidations.

### 3.2. Pulmonary Function Tests

PFTs are summarized in Table 2. On average pulmonary function values were within the normal range for all the groups, although in Group A, FVC, and FEV_1_ were slightly smaller (FEV_1_ Group A: 99.8%pr ± 13.5 vs. FEV1 Group B 104.3%pr ± 12.6; *p* = 0.04 and FVC Group A: 100.8%pr ± 14.8 vs. FVC Group B 107.4%pr ± 12.1; *p* =0.003). Similarly, DL’co was within the normal range but smaller in group A (DL’co Group A: 89.6%pr ± 15.3 vs. DL’co Group B 92.4%pr ± 13.4; *p* = 0.023). The DL’co/VA was similar between groups (DL’co Group A: 98.9%pr ± 18.1 vs. DL’co Group B: 98.1%pr ± 19.5; *p* = 0.806).

### 3.3. Symptoms

Table 3 describes the symptoms of the studied population. Fever (58%), cough (40%), fatigue (30%), and dyspnea (27%) were the symptoms most described by the entire population. Between the two populations, there was a significant difference in the frequency distribution of symptoms (*p* < 0.004). In Group A, the most common symptoms were fever (48%) and dyspnea (37%), followed by cough (28%) and fatigue (28%), in group B instead the most common symptoms were fever (69%) and cough (53%) followed by fatigue (32%) and headache (24%).

### 3.4. Exercise Tests

MWT, HG, and 1′MSTS are shown in Table 2 and Figure 1.

Six subjects did not perform the HGT (4 in Group A and 2 in Group B). In the 6MWT, Group A walked a significantly shorter distance with a lower nadir saturation compared to Group B (6MWT distance Group A: 549 m ± 115 vs. Group B 606 m ± 90; *p* = 0.001).

Thirty-two subjects did not perform the 1′STST (28 in Group A and 4 in Group B). In the 1′STST, Group A performed fewer repetitions with a lower nadir saturation than Group B (1′STS repetition Group A: 32.3 ± 8.5 vs. Group B 35.4 ± 8.9; *p* = 0.05).

Eleven subjects did not perform the HGT (8 in Group A and 3 in Group B). No differences between groups were observed at the HG test.

### 3.5. Correlations

Regression analysis is shown in Figure 1. The 6MWT distance correlated with FVC, %pr (R^2^ = 0.059, *p* = 0.005) and DL’co, %pr (R^2^ = 0.107, *p* < 0.001), while it showed no correlation with FEV_1_, %pr (R^2^ = 0.014, *p* = 0.177). The number of repetitions correlated with FVC, %pr (R^2^ = 0.078, *p*. = 0.002) and DL’co, %pr (R^2^ = 0.086, *p* = 0.001), while it showed no correlation with FEV_1_, %pr (R^2^ = 0.032, *p* = 0.052). The strength at the HGT correlated only with DL’co, %pr (R^2^ = 0.08, *p* < 0.001), while it showed no correlation with FVC, %pr (R^2^ = 0.002, *p* = 0.582) and FEV_1_, %pr (R^2^ = 0.000, *p* = 0.799).

## 4. Discussion

These data from our study confirm that in the majority of subjects, the impairment of respiratory function, even if minimal, has an impact on exercise tolerance, presenting in the last few waves a lower impairment of respiratory function and stress tests. This is the first study that has shown that those infected in recent waves and likely with the omicron variant have less impairment in lung function and exercise tolerance.

As SARS-CoV-2 evolves by acquiring mutations, clinical phenotypes are also changing. Previous studies have shown that in post-COVID-19 survivors, the main pulmonary function abnormalities were a reduction in DL’co with an almost normal DL’co/VA and a reduction in FVC and TLC with an increase in the FEV_1_/FVC ratio [12,13,14,29,30,31,32,33]. Our group has already demonstrated that about six months after COVID, PFTs were mostly normal, with a slight difference in patients with a more severe clinical presentation of the disease, where the FVC and TLC were statistically significantly reduced, although clinically within the normal range [16,17]. The present study confirms our previous findings even in a highly selected category of subjects such as healthy and fit military personnel.

Pulmonary function tests were within the normal range; nevertheless, in Group A, the FEV_1_, FVC, and DL’co, but not the DL’co/VA, were slightly reduced compared to Group B (Table 2). So, this study, compared to previous studies, adds another piece of information, which is that impairment of lung function decreases with the mutation of viral infection. Two datapoints seem to support this hypothesis, the lower lung damage of group B (in this group, the pulmonary impairment is negligible) and the lower need for hospitalization.

There are not many studies exploring the effect of the different variants on clinical presentations, and as in our study, the SARS-CoV-2 variant was inferred from the period of infection. For example, a recent study demonstrated that liver injury is more common in patients infected with Delta variants [34], and the studied patients were divided into “pre-Delta period” if they were infected between 1 February and 30 November 2020 or in the Delta period between 1 August and 31 August 2021. Yoon et al. [35] compared the pulmonary involvement at the Chest CT scan in 88 patients with the Omicron variant to 88 patients with the Delta variant and demonstrated that in hospitalized patients with similar disease severity, the Omicron SARS-CoV-2 variant showed nontypical peribronchovascular pneumonia and less pulmonary vascular involvement than the Delta variant. In addition, in this case the variant of the virus was inferred from the period of infection, as CT images were collected in November 2021 (Delta variant) and February 2022 (Omicron variant).

Several studies have demonstrated a reduction in DL’co in COVID patients [36,37]. DL’co may be lower in COVID-19 patients due to a combination of lung inflammation, pulmonary fibrosis, blood clotting, and ventilation-perfusion mismatch [12]. In our study, although within the normal limits, DL’co was reduced than DL’co/VA. Likely, this was related to an impairment of the diffusion linked mainly to the reduction in alveolar volume [12]. Barisione et al. demonstrated that in subjects recovering from COVID-19 pneumonia, lung diffusion of nitric oxide, which is more specific to detect lung diffusion abnormalities, independently from the pulmonary capillary blood volume, was more frequently impaired, confirming that there mainly is alveolar-capillary damage and loss of alveolar units with capillary volume relatively preserved [38].

However, this study shows that even small variations of DL’co in healthy, fit subjects can cause a limitation in exercise tolerance, that in athletes or soldiers can have a major clinical impact. Previously we have already demonstrated that in COVID survivors, there was a desaturation during the 6MWT and that the DL’co correlated with the desaturation distance ratio at the 6MWT, which is an index to assess the involvement of the alveolar-capillary membrane [16]. In the current study, we did not demonstrate any desaturation during the 6MWT, but the DL’co was the best measurement to correlate with 6MWT distance, HGT forces, and 1′STST repetition (Figure 1), all measurements of exercise tolerance and muscle strength.

DL’co is a measure of the lung’s ability to transfer gas from the alveoli into the blood and depends on several factors: alveolar surface area, the thickness of the alveolar-capillary membrane, hemoglobin concentration, ventilation-perfusion mismatch, lung volumes [12,39]. Exercise increases the demand on the lungs to transfer gases across the alveolar-capillary membrane. In this sense, a small decreased DL’co at rest could indicate that the lung is less able to transfer gas efficiently during exercise. In fact, a common finding from the cardiopulmonary exercise test in post-COVID subjects is a ventilatory inefficiency suggesting an abnormal response to carbon dioxide production [40].

This study has also shown that there is a different distribution of the presentation of symptoms depending on the period of infection and, therefore, on the different variants. Fever is still the most common symptom in the whole population, but the group of subjects infected before September 2021 presented a higher frequency of dyspnea and a lower frequency of cough (Table 3). On the other hand, Group B, infected with the most recent variants, showed symptoms of infection more typical of the upper respiratory tract (sore throat or cough) or general symptoms such as headache, fatigue, or joint/muscle pain. This difference may be due to several reasons. First of all, a different tropism of the variants. Hui et al. compared the replication competence and cellular tropism of Alpha, Beta, Delta, and Omicron variants in ex vivo explant cultures of human bronchi and lungs [41] and demonstrated that Omicron replicates faster than all other SARS-CoV-2 variants in the bronchi but less efficiently in the lung parenchyma. This could explain the reduced severity of Omicron and the difference in symptoms in Group B compared to Group A. Moreover, different immunological and vaccination statuses may have affected the response to the virus and clinical presentation.

The main limitation of this study is that the analysis of the SARS-CoV-2 variants has not been carried out, and it is not possible to affirm with confidence that two populations were infected by different variants. Nevertheless, the Italian epidemiological data [3] seems to strongly support the hypothesis that in the first group, the beta and gamma variants were predominant, while in the second group, the omicron variant was predominant. Other limitations of the present study are mainly related to its retrospective nature. There is an age difference between the two groups. This could either refer to a change in the ability of the virus to infect younger people or a bias in our data. Moreover, the follow-up differs between groups, although it was longer in Group A with the worst PFTs, that which could suggest an underestimation of their values and so a bigger difference between groups. Some data are missing, mainly on HGT, as well as the possibility of comparing pulmonary function tests before SARS-CoV-2 infection for an assessment of the real impact of infection on lung function [34]. Moreover, the Chest CT scan was not performed for all subjects, and a qualitative analysis was performed.

## 5. Conclusions

This study shows that the SARS-CoV-2 infection in healthy middle-aged military outpatients was more severe in the first waves than in the later ones and that, in healthy and physically fit individuals, even a marginal reduction in resting respiratory test values can have a major impact on exercise tolerance and muscles strength. Despite it being difficult to know which is the determining factor, the combination of the vaccination strategy, the change in the virus virulence, or differences in the management, the latest waves of infection have been characterized by more symptoms of the upper respiratory tract and less lung involvement compared to the previous ones.

## Figures and Tables

**Figure 1 diagnostics-13-01679-f001:**
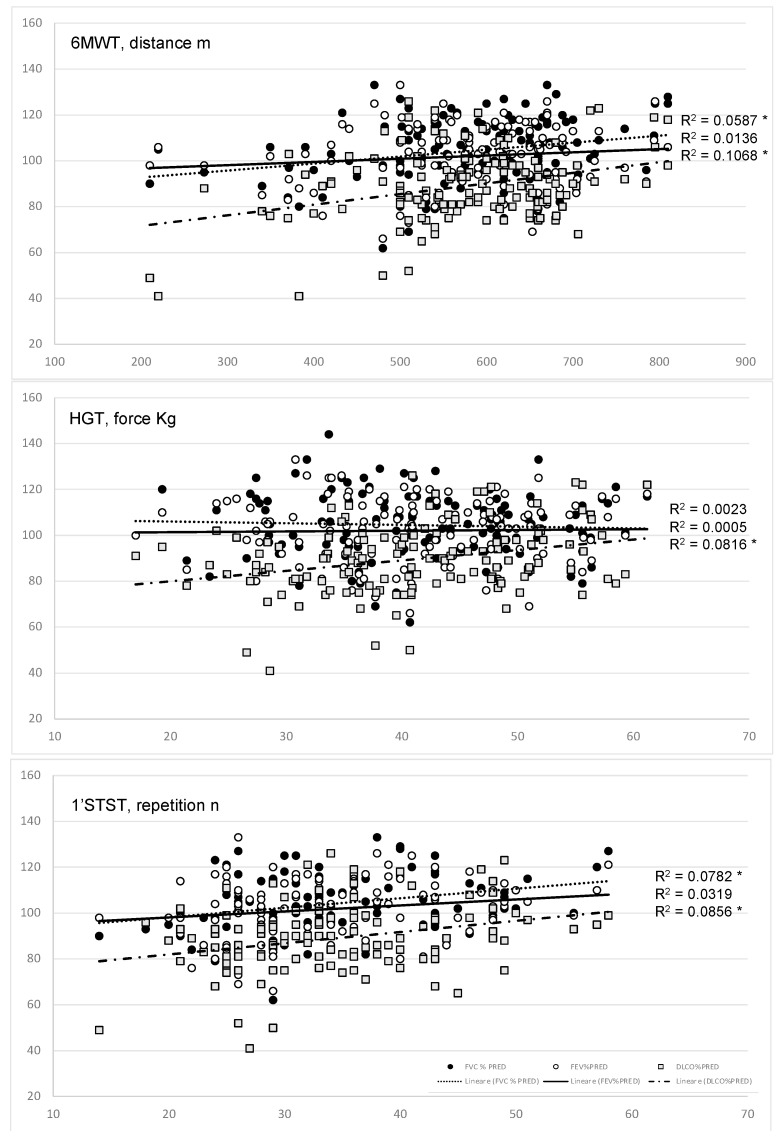
Correlations of FVC (black circles), FEV_1_ (white circles), and DL’co (grey squares) with the 6MWT distance on the top, the HGT force in the middle, and 1‘STST number of repetitions on the bottom. The DL’co significantly correlates with all the measures, FVC just with the 6MWT distance and 1′STST repetitions. FEV1 does not correlate with any measurement. 1′STST: one minute sit to stand test; 6MWT: six minute walk test; HGT: Handgrip Test; DL’co: diffusing lung capacity for carbon monoxide, FEV_1_: forced Expiratory Volume in the first second; FVC: Forced Vital Capacity. * means *p* < 0.01.

**Table 1 diagnostics-13-01679-t001:** Subjects’ characteristics.

	All Group (*n* = 153)	Group A (*n* = 79)	Group B (*n* = 74)	*p*-Value
Female/male, *n*	12/141	4/75	8/66	ns
Age, yrs	47.0 ± 9.8	49.6 ± 9.8	44.2 ± 9.1	0.001
Height, cm	169.8 ± 28.4	166.8 ± 38.8	173.0 ± 6.4	ns
Weight, kg	85.2 ± 15.3	88.4 ± 14.8	81.8 ± 15.2	0.007
BMI, kg/m^2^	28.1 ± 4.4	28.6 ± 4.4	27.5 ± 4.3	0.095
Follow-up, days	140 ± 88	159 ± 105	119 ± 66	0.006
Immunization status: - 0 dose, *n* (%) - 1 dose, *n* (%) - 2 or more doses, *n* (%) - Previous infection, *n* (%)	83 (54.2) 4 (2.6) 66 (43.1) 11 (7.1)	76 (96.2) 3 (3.8) 0 (0.0) 0 (0.0)	7 (9.5) 1 (1.4) 66 (89.2) 11 (14.9)	<0.0001
Hospitalization, *n* (%)	66 (43.1)	57 (72.2)	9 (12.2)	<0.0001
Oxygen therapy, *n* (%)	32 (20.9)	28 (35.4)	4 (5.4)	<0.0001
High-flow oxygen therapy or non-invasive ventilation	8 (5.2)	7 (8.9)	1 (1.4)	ns
SAH, *n* (%)	26 (17.0)	22 (27.8)	4 (5.4)	0.005
IHD, *n* (%)	3 (2.0)	2 (2.5)	1 (1.4)	ns
Metabolic disorders, *n* (%) - hypercholesterolemia, *n* (%) - Type 1 diabetes, *n* (%) - Type 2 diabetes, *n* (%)	43 (28.1) 33 (21.6) 1 (0.7) 9 (5.9)	14 (17.7) 8 (10.1) 1 (1.3) 5 (6.3)	29 (39.2) 25 (33.8) 0 (0.0) 4 (5.4)	0.005
asthma, *n* (%)	11 (7.2)	5 (6.3)	6 (8.1)	ns
Therapy: - Heparin, *n* (%) - Antibiotics, *n* (%) - Oral Corticosteroids, *n* (%) - Remdesevir, *n* (%) - Tocilizumab, *n* (%)	44 (28.8) 66 (43.1) 51 (33.3) 26 (17.0) 5 (3.3)	42 (53.2) 44 (55.7) 39 (49.4) 25 (31.2) 5 (6.3)	4 (5.4) 22 (29.7) 12 (16.2) 1 (1.4) 0 (0.0)	<0.01

BMI: Body Mass Index; IHD: Ischemic Heart Disease; SAH: Systemic Arterial Hypertension. Values are expressed as mean ± SD.

**Table 2 diagnostics-13-01679-t002:** Subjects’ pulmonary function tests and exercise tests (six min walk test, hand grip, and 1 min sit to stand test).

	All Group (*n* = 153)	Group A (*n* = 79)	Group B (*n* = 74)	*p*-Value
FEV_1_, l (FEV_1_, %pr)	3.67 ± 0.62 (102.0 ± 13.2)	3.61 ± 0.65 (99.8 ± 13.5)	3.75 ± 0.57 (104.3 ± 12.6)	ns 0.04
FVC, l (FVC, %pr)	4.60 ± 0.76 (104.1 ± 13.9)	4.51 ± 0.82 (100.8 ± 14.8)	4.70 ± 0.69 (107.4 ± 12.1)	ns 0.003
FEV1/FVC, %	80.2 ± 6.2	80.2 ± 5.5	80.2 ± 7.0	ns
PEF l/min,	9.8 ± 1.6	9.8 ± 1.6	9.8 ± 1.7	ns
DL’co-sb, %pr	89.6 ± 15.3	86.7 ± 16.6	92.4 ± 13.4	0.023
DL’co-sb/VA, %pr	98.5 ± 18.8	98.9 ± 18.1	98.1 ± 19.5	ns
6MWTD, m	577 ± 107	549 ± 115	606 ± 90	0.001
SpO_2_ rest, %	97.1 ± 1.4	96.9 ± 1.6	97.2 ± 1.1	ns
SpO_2_ nadir, %	95.7 ± 1.6	95.1 ± 1.8	96.3 ± 1.0	0.001
HR rest, bpm	82.9 ± 14.1	82.6 ± 14.4	83.2 ± 13.9	ns
HR max, bpm	124.6 ± 17.7	113.1 ± 14.4	117.0 ± 15.1	ns
HG right hand, kg	40.8 ± 9.5	41.6 ± 10.3	40.1 ± 8.6	ns
HG right hand, kg	39.0 ± 9.6	39.3 ± 8.8	38.7 ± 10.3	ns
1′MSTS, *n* repetition	34.1 ± 8.8	32.3 ± 8.5	35.4 ± 8.9	0.05
SpO_2_ rest, %	97.6 ± 1.5	97.2 ± 1.5	97.8 ± 1.4	0.035
SpO_2_ Nadir, %	96.9 ± 2.5	96.1 ± 3.5	97.4 ± 1.2	0.003
HR rest, bpm	85.1 ± 16.0	89.0 ± 17.7	82.3 ± 14.1	0.023
HR max, bpm	123.5 ± 18.7	122.8 ± 21.1	123.9 ± 17.0	ns

%pr: %predicted value, Values are expressed as mean ± SD. 1′MSTS: 1 min sit to stand; 6MWTD: six-minute walk test distance; DLco-SB: diffusing lung capacity for carbon monoxide—single breath; FEV_1_: Forced Expiratory Volume in 1 s; FVC: forced vital capacity; HG: hand grip; HR: Heart Rate; SpO_2_: Peripheral saturation of oxygen; VA: Alveolar Ventilation.

**Table 3 diagnostics-13-01679-t003:** Subjects’ symptoms at the time of SARS-CoV-2 infection. The distribution of symptoms in the two population is significantly different (Test Chi Square, *p* < 0.004).

	All Group (*n* = 153)	Group A (*n* = 79)	Group B (*n* = 74)
Fever, *n* (%)	89 (58)	38 (48)	51 (69)
Cough, *n* (%)	61 (40)	22 (28)	39 (53)
Fatigue, *n* (%)	46 (30)	22 (28)	22 (32)
Dyspnea, *n* (%)	41 (27)	29 (37)	12 (16)
Headache, *n* (%)	26 (17)	8 (10)	18 (24)
Loss of taste, *n* (%)	26 (17)	14 (18)	12 (16)
Loss of smell, *n* (%)	23 (15)	14 (18)	9 (12)
Arthralgia/myalgia, *n* (%)	25 (16)	9 (11)	16 (22)
Sore throat, *n* (%)	18 (12)	4 (5)	14 (19)
Other symptoms, *n* (%)	25 (16)	14 (18)	11 (15)

## Data Availability

The data presented in this study are available on request from the corresponding author.

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
