# Peer review of "Effects of SARS-CoV-2 on Pulmonary Function and Muscle Strength Testing in Military Subjects According to the Period of Infection: Cross-Sectional Study"

_diagnostics, 2023, doi:10.3390/diagnostics13101679_

Round 1
Reviewer 1 Report
Dear Sir/ Madam,
Overall the manuscript is well prepared and interesting to read. But, at the authors pointed, there are some limitations in this study; perhaps the major concern for me is not knowing the volunteers were affected by which variants of the virus. Additionally, it would be more interesting if chest CT scan or at least X-Ray data of the participants were recorded alongside of other testes to be correlated with long term respiratory impairments as a post covid infection.
Author Response
Response to reviewer 1
We thank the reviewer for reviewing our manuscript and for helpful comments
Reviewer 1
R1: Overall the manuscript is well prepared and interesting to read. But, at the authors pointed, there are some limitations in this study; perhaps the major concern for me is not knowing the volunteers were affected by which variants of the virus.
Reply: we completely agree with the reviewer, but unfortunately we do not have the genetic sequencing of the virus. In reference 3, although it is in Italian language, it has shown that the gamma variant finished mostly in July and the omicron started in November (as shown in the figure). In the first group (Group A) just three subjects were infected between May and July and in the second group (Group B) the first infected subjects were observed in November, so we could confidently say that in the two groups the variants were different, probably mostly beta and gamma in Group A and omicron in group B (https://www.epicentro.iss.it/coronavirus/pdf/sars-cov-2-monitoraggio-varianti-rapporti-periodici-31-marzo-2023.pdf) (see figure below)
We have amended the manuscript in the study limitations section and softened the conclusions.
R1: Additionally, it would be more interesting if chest CT scan or at least X-Ray data of the participants were recorded alongside of other testes to be correlated with long term respiratory impairments as a post covid infection.
Reply: We added the data of the CT scan in the results.
Reviewer 2 Report
I would like to thank the editors of the Diagnostics for this opportunity provided for me to review the manuscript entitled “Effects of SARS-CoV-2 on pulmonary function and muscle strength testing in military subjects according to the period of infection: a cross-sectional study”. In this study of 153 post-COVID-19 patients, the authors compared the functional outcomes of two groups of patients infected during the different waves of the disease. The investigation addresses an important topic regarding the long-term consequences of COVID-19. Few studies have previously been conducted on this topic and this study can add additional insights concerning the sequela of various variants of SARS-CoV-2. However, several crucial points merit consideration prior to publication:
1. As noted in Table 1, there are substantial differences between the two groups with respect to demographics and comorbidities. For instance, patients in group A were approximately 5 years older than the patients in group B. These baseline differences, at least to some extent, may have resulted in differences in pulmonary function tests and functional measures. Therefore, the results need to be adjusted for these baseline features.
2. The authors did not provide sufficient data (e.g., the results of chest CT, mechanical ventilation, ICU admission, etc.) regarding the severity of the disease in the two groups of patients. The data regarding the severity of the disease needs to be added to the manuscript.
3. Vaccination and immunization history are vital aspects in assessing the severity and long-term complications of COVID-19. The data regarding vaccination history and prior infection with SARS-CoV-2 should be added to the study to shed light on the findings of the study.
4. If available, I suggest the authors add additional information regarding the therapeutic measures for the included patients. Special attention needs to be given to anticoagulation therapy, as thromboembolic complications could have resulted in alterations in pulmonary function tests and other functional outcomes.
Author Response
Response to Reviewer 2
We thank the reviewer for reviewing our manuscript and for helpful comments
R2: I would like to thank the editors of the Diagnostics for this opportunity provided for me to review the manuscript entitled “Effects of SARS-CoV-2 on pulmonary function and muscle strength testing in military subjects according to the period of infection: a cross-sectional study”. In this study of 153 post-COVID-19 patients, the authors compared the functional outcomes of two groups of patients infected during the different waves of the disease. The investigation addresses an important topic regarding the long-term consequences of COVID-19. Few studies have previously been conducted on this topic and this study can add additional insights concerning the sequela of various variants of SARS-CoV-2.
Reply: We thank the reviewer.
However, several crucial points merit consideration prior to publication:
R2: 1. As noted in Table 1, there are substantial differences between the two groups with respect to demographics and comorbidities. For instance, patients in group A were approximately 5 years older than the patients in group B. These baseline differences, at least to some extent, may have resulted in differences in pulmonary function tests and functional measures. Therefore, the results need to be adjusted for these baseline features.
Reply: We agree with the reviewer that there is a difference in age and a not significant difference in BMI, but this is a retrospective study and we have included all the subjects (except those with missing data), so this results probably refers to the target population of the variants or to the other characteristics of the virus and population. On the other hand to uniform the two groups would mean to arbitrarily take out some subjects from the first group and this could be generate other bias. We would like to point out that pulmonary function tests are presented as absolute values and in percent of the predicted just to normalized the data by age, weight, ethnics and gender that is exactly what the reviewer is asking for. However, we have added a sentence in the limitation paragraph.
R2: 2. The authors did not provide sufficient data (e.g., the results of chest CT, mechanical ventilation, ICU admission, etc.) regarding the severity of the disease in the two groups of patients. The data regarding the severity of the disease needs to be added to the manuscript.
Reply: We thank the reviewer. We amended adding the data in the table and in the results.
R2: 3. Vaccination and immunization history are vital aspects in assessing the severity and long-term complications of COVID-19. The data regarding vaccination history and prior infection with SARS-CoV-2 should be added to the study to shed light on the findings of the study.
Reply: We thank the reviewer. We amended adding the data in the table and in the results
R2: 4. If available, I suggest the authors add additional information regarding the therapeutic measures for the included patients. Special attention needs to be given to anticoagulation therapy, as thromboembolic complications could have resulted in alterations in pulmonary function tests and other functional outcomes.
Reply: We thank the reviewer. We amended adding the data in the table and in the results
Reviewer 3 Report
The subject you present is interesting, but I have some doubts, which I will list below.
1. Why table 1 shows only the % of female participants? Any reason?
2. In the same table, the values presented for clinical variables such as Ischemic Heart Disease and Systemic Arterial Hypertension are relatively low percentages. What is the time of evolution of these diseases? Does it correspond only to women?
3. I believe that Table 2 should be re-analyzed because the results of pulmonary function tests could be biased by age, sex, physical condition, comorbidity, etc., together with the time of recovery from SARS infection.
4. Since the work does not present results of molecular tests to prove the circulating SARS strain and is only based on the temporality of the cases, I consider that the conclusion will have to be careful and not say coincidentally that the last variants of SARS-CoV-2 cause less respiratory disorders than the first ones, I think they may correspond to the fact that a large percentage of the population was already vaccinated, have you considered this?
Author Response
Response to Reviewer 3
We thank the reviewer for reviewing our manuscript and for helpful comments
R3: The subject you present is interesting, but I have some doubts, which I will list below.
- Why table 1 shows only the % of female participants? Any reason?
Reply: The first row of the table indicated the number of women and the percentage of women in the whole group, while the table refers to all subjects. However, to avoid any possible confusion, we have changed the row of the table in female/male to make it more understandable.
R3: 2. In the same table, the values presented for clinical variables such as Ischemic Heart Disease and Systemic Arterial Hypertension are relatively low percentages. What is the time of evolution of these diseases? Does it correspond only to women?
Reply: The data refer to the entire population. The low percentages of the diseases are related to the type of population that is of healthy soldiers. We do not have the exact time of recovery, also because the recovery was differently evaluated according to the severity of the infection (negative swab) or according to the law (5-15 days according to the month and year). We do not have the exact moment of the recovery, also because the healing (or negativity from infection) was evaluated differently depending on the severity of the subject, in the most serious ones several swabs were made until negativity especially for the subjects of the first waves, while in the less serious ones the negativity was assumed after 5 days of the end of symptoms according to the national and international recommendation.
R3: 3. I believe that Table 2 should be re-analyzed because the results of pulmonary function tests could be biased by age, sex, physical condition, comorbidity, etc., together with the time of recovery from SARS infection.
Reply: The majority of study have analyzed the time from the infection (positive swab) and as well the definition of long COVID is based on the time of the positive swab, because it is impossible to certify for all subjects the time of recovery. In addition, some authors may object that in most patients healing occurs before the negativity of the swab (as clinical or asymptomatic for example) and in others much later as in patients. with post-long COVID. Looking at the follow up, the group A did the functional tests on average later than the group B and so PFTs and theoretically pulmonary function tests should have been better (longer recovery) than group B, but actually it was the opposite, further confirming our data. About the normalization of data see the answer to reviewer 2.
R3: 4. Since the work does not present results of molecular tests to prove the circulating SARS strain and is only based on the temporality of the cases, I consider that the conclusion will have to be careful and not say coincidentally that the last variants of SARS-CoV-2 cause less respiratory disorders than the first ones, I think they may correspond to the fact that a large percentage of the population was already vaccinated, have you considered this?
Reply: thank you we have modified the conclusion according to this and the other reviewers and added the data of vaccination.
Reviewer 4 Report
The paper submitted for review focuses on a very important and contemporary problem of the protracting post-COVID dysfunctions of the respiratory system. Please find the comments below: 1. SARS-CoV-2 not SARS-CoV2
2. In the introduction, the order of the paragraphs appears to be wrong. The penultimate one: "Symptoms of COVID-19 can appear 2-14 days after exposure..." should come before the reference to complications and long-Covid.
In my opinion, rather than listing the symptoms of Covid, which could be omitted altogether at this point, authors should provide information on long-Covid - esp. it’s definition and most common symptoms.
While there are many reports on the prevalence of long-covid symptoms, the authors cited only one of them - ref. no 8. It should be changed. 3. Material and method
In this section, please specifically indicate what was the initial number of test subjects, how many were rejected and for what reason (e.g. by flowchart) . Some information on this is mentioned in "results". There is no information on the time criterion for both groups (exactly from when and until when) - the above was mentioned in the abstract but not in the main text. The same goes for the information about the negative smear test after 3 months.
4. Results: The analysis of "metabolic disorders" should be further expanded. They appear to be significantly more frequent in the study group. Could this impact the results?
Shouldn't patients with asthma be removed from the analysis? Table 3 lists symptoms, but it is not clear whether these are symptoms of covid or long-covid. If these are for Covid, there should be further information on symptoms of long COVID.
The symptomatology section should be placed before the test results. Since the tests were done on average 6 months after the infection, the authors should use a term "after about 6 months" in the discussion. The lack of a definite, single time point is also a limitation.
Author Response
Response to Reviewer 4
We thank the reviewer for reviewing our manuscript and for helpful comments
R4: The paper submitted for review focuses on a very important and contemporary problem of the protracting post-COVID dysfunctions of the respiratory system. Please find the comments below:
R4: 1. SARS-CoV-2 not SARS-CoV2
Reply: We thank the reviewer. We amended the manuscript accordingly.
R4: 2. In the introduction, the order of the paragraphs appears to be wrong. The penultimate one: "Symptoms of COVID-19 can appear 2-14 days after exposure..." should come before the reference to complications and long-Covid.
Reply: We thank the reviewer. We amended the manuscript accordingly.
R4: In my opinion, rather than listing the symptoms of Covid, which could be omitted altogether at this point, authors should provide information on long-Covid - esp. it’s definition and most common symptoms.
Reply: thank you for your comment, we added symptoms of the COVID because we reported symptoms of the COVID in our population. We added the definition and symptoms of the long COVID too.
R4: While there are many reports on the prevalence of long-covid symptoms, the authors cited only one of them - ref. no 8. It should be changed.
Reply: We thank the reviewer. We amended the manuscript accordingly.
R4: 3. Material and method
In this section, please specifically indicate what was the initial number of test subjects, how many were rejected and for what reason (e.g. by flowchart) . Some information on this is mentioned in "results". There is no information on the time criterion for both groups (exactly from when and until when) - the above was mentioned in the abstract but not in the main text. The same goes for the information about the negative smear test after 3 months.
Reply: thank you, we amended it. We added the number of subjects evaluated (170) in the results section and the reason why they were excluded by the analysis (mostly because of the lack of follow up). There was not an exact criteria for the follow up on average it was after three months but it depended on the clinical status of the subject. In addition, it is a retrospective study and the management of covid has been different from the timing and knowledge of the contagion. There was a mistake in the abstract, we apologize for the error and amended it.
R4: 4. Results: The analysis of "metabolic disorders" should be further expanded. They appear to be significantly more frequent in the study group. Could this impact the results?
Reply: This group of subjects are generally healthy subjects, soldiers fit for the service, out of 43 subjects with metabolic disorders 33 had just hypercholesterolemia and the other 10 diabetes (one of them Type 1 diabetes). We better clarify this point in the table and in the result.
R4: Shouldn't patients with asthma be removed from the analysis?
Reply: They were well balanced between groups and study on the effect of asthma and asthma therapy on COVID are controversial. Anyways, to remove them out does not affect the results.
R4: Table 3 lists symptoms, but it is not clear whether these are symptoms of covid or long-covid. If these are for Covid, there should be further information on symptoms of long COVID.
Reply: we amended it, they are symptoms of the COVID (not post COVID) we have the post covid symptoms mostly for all the group B subjects but just for few group A subjects. We add a sentence in the limitation.
R4: The symptomatology section should be placed before the test results. Since the tests were done on average 6 months after the infection, the authors should use a term "after about 6 months" in the discussion. The lack of a definite, single time point is also a limitation.
Reply: we amended the manuscript including in the limitation.
Round 2
Reviewer 1 Report
Dear authors,,
You have made a good improvement on the manuscript. However, It would have been better if you added an example of the chest CT scan in your manuscript for each group A, B, and C.
Reviewer 2 Report
In my opinion, the authors have sufficiently addressed the comments of reviewers. In addition, the authors have added crucial data requested by the reviewers in previous round of review. I have no other comments or concerns regarding this study.
Reviewer 3 Report
The authors have responded to my comments and I recommend publication.
Reviewer 4 Report
none